# Mortality among HIV-Positive and HIV-Negative People Who Inject Drugs in Mizoram, Northeast India: A Repeated Cross-Sectional Study (2007–2021)

**DOI:** 10.3390/ijerph21070874

**Published:** 2024-07-04

**Authors:** Lucy Ngaihbanglovi Pachuau, Caterina Tannous, Richard Lalramhluna Chawngthu, Kingsley Emwinyore Agho

**Affiliations:** 1School of Health Sciences, Western Sydney University, Campbelltown, NSW 2560, Australia; c.tannous@westernsydney.edu.au (C.T.); k.agho@westernsydney.edu.au (K.E.A.); 2Mizoram State Aids Control Society, Aizawl 796001, Mizoram, India; docchongthu@gmail.com; 3Translational Health Research Institute (THRI), Western Sydney University, Campbelltown, NSW 2560, Australia; 4African Vision Research Institute (AVRI), Westville Campus, University of KwaZulu-Natal, Durban 3629, South Africa

**Keywords:** human immunodeficiency virus, injecting drug user, mortality rate, trend, India

## Abstract

**Background:** HIV and drug overdose continue to be the leading causes of death among people who inject drugs (PWID). Mizoram, a small state in the northeast of India, has the highest prevalence of HIV in India and a high HIV prevalence among PWID. **Objective:** To estimate the mortality among HIV-positive and HIV-negative PWID and to describe its associated factors. **Methods:** Cross-sectional datasets from the 2007–2021 Mizoram State AIDS Control Society (MSACS) data comprising 14626 PWID were analyzed. Logistic regression analysis was conducted to examine the factors associated with mortality among HIV-negative and HIV-positive PWID after adjusting for potential confounding factors. **Results:** Mortality among HIV-negative PWID declined by 59% between 2007 and 2021. The mortality rate among HIV-positive PWID also declined by 41% between 2007 and 2021. The multiple logistic regression analysis revealed that being divorced/separated/widowed (AOR = 1.41, 95% CI 1.03–1.94) remained positively associated with mortality among HIV-positive PWID. Mortality among HIV-negative PWID remained positively associated with ages of 24–34 years (AOR = 1.54, 95% CI 1.29–1.84) and above 35 years (AOR = 2.08, 95% CI 1.52–2.86), being divorced/separated/widowed (AOR = 1.28, 95% CI 1.02–1.61), and the sharing of needles/syringes (AOR = 1.28, 95% CI 1.34–2.00). Mortality among HIV-negative PWID was negatively associated with being married (AOR = 0.72, 95% CI 0.57–0.90), being employed (AOR = 0.77, 95% CI 0.64–0.94), and having a monthly income. **Conclusions:** The mortality rate among HIV-negative and HIV-positive PWID declined significantly between 2007 and 2021 in Mizoram. To further reduce mortality among PWID, interventions should target those sharing needles/syringes, those above 24 years of age, and unmarried participants.

## 1. Introduction

PWID are at increased risk of premature death [1,2]. Globally, the leading causes of death among PWID are accidental drug overdose and human immunodeficiency virus (HIV) infection. A systematic review and meta-analysis conducted on 67 cohort studies in 2013 estimated that PWID had a crude mortality rate of 2.35 deaths per 100 person-years, a rate that is 14.7 times higher than the general population [1,3]. This study also found that the mortality was three times higher among HIV-positive PWID compared to HIV-negative PWID.

ALIVE (AIDS Linked to the Intravenous Experience), a long-standing community-based prospective study in Baltimore which followed PWID for 30 years (1988–2018), found that more than 40 percent of their participants died during follow-up (median = 13 years), primarily from HIV/AIDS, drug overdose, and chronic diseases [4]. They also found that HIV/AIDS-related deaths declined after 1997 following the widespread availability of combination antiretroviral therapy (cART). However, the same study reported that drug-related deaths among the participants increased exponentially, more than 80 times the national average, and were probably driven by the non-medical use of prescriptions and easy availability of fentanyl [4]. A similar prospective cohort study conducted in Vancouver, Canada, also found an increased risk of death among HIV-positive PWID compared to HIV-negative PWID. However, in contrast to the ALIVE study, they found declining overdose deaths among the participants and suggested that this may be related to improved harm reduction strategies in Vancouver [5].

In 2021, the Joint United Nations Programme on HIV and AIDS (UNAIDS) reported that the risk of acquiring HIV is 35 times higher among people who inject drugs [6]. However, with the increased availability of cART, people infected with HIV have enjoyed substantial reductions in HIV/AIDS-associated morbidity and mortality [7,8]. Highly active antiretroviral treatment (HAART) has been shown to improve the course of HIV disease and subsequently decrease the mortality rate among all HIV-infected populations, including PWID [5,9]. However, HIV-positive PWID are often less likely to benefit from treatment due to less-than-optimal adherence to treatment, leading to a high mortality rate among this population [9].

The highest prevalence of HIV in India is found among PWID [10]. However, few reports of mortality among PWID are available. A longitudinal cohort study conducted in Chennai, South India, reported that the mortality among PWID was 4.3 per 100 person-years [11]. The study also found that HIV-positive PWID who were not immunosuppressed at baseline had mortality rates comparable to HIV-negative PWID, suggesting that good adherence to highly active antiretroviral treatment (HAART) by this population would substantially impact mortality [11]. 

In India, the AIDS-related deaths per 100,000 declined from 15.04 in 2010 to 3.08 in 2021 [12]. The Indian government rolled out free antiretroviral treatment (ART) under the national program in April 2004; this has led to a decline in AIDS-related deaths in the last several years [10]. Mizoram, a state in the northeast of India, has also seen a decline in AIDS-related deaths from 58.71 deaths per 100,000 in 2010 to 15.80 in 2021. The decline in AIDS-related deaths in Mizoram could be attributed to the introduction of ART in 2007. To further combat AIDS-related deaths in Mizoram, there has been a scale-up of ART centers in different districts across the state, with increased efforts to link people living with HIV to ART services [13]. However, despite these efforts, the number of AIDS-related deaths in Mizoram is still five times higher than the national average.

Despite the high number of AIDS-related deaths reported in Mizoram, no research has been conducted on the mortality rate among PWID in this region. Hence, the current study aimed to estimate the mortality rate among PWID over a period of 15 years and to determine the associated factors of mortality among HIV-positive and HIV-negative PWID in Mizoram, India. The findings from this study would enable health administrators, public health researchers, and government policymakers to reassess and improve the current intervention strategies aimed at reducing HIV-related deaths among people who inject drugs in Mizoram.

## 2. Materials and Methods

### 2.1. Study Sample and Design

This was a cross-sectional study and used secondary data on PWID who were registered in targeted intervention (TI) services under the Mizoram State AIDS Control Society (MSACS). The secondary data were from MSACS, and these datasets were accessed on 1 April 2021. The datasets used for this study can be accessed upon request. MSACS is an organization created by the Government of Mizoram on behalf of the state to respond to the HIV/AIDS epidemic and to deliver effective and efficient implementation of the AIDS control program. Datasets were extracted from TI-registered PWID. Targeted intervention (TI) is one of the many core strategies for HIV prevention among PWID [14]. Harm reduction strategies under TI in Mizoram focus on major components including behavior change communication, treatment of sexually transmitted infections, distribution of condoms and other risk reduction materials, needle exchange programs, and opioid substitution therapy [13]. Datasets from January 2007 to January 2021 were used to estimate the mortality rate among HIV-positive and HIV- negative PWID. A total of 14681 PWID were registered in the TI services between January 2007 and January 2021. ART was first introduced in 2007 in Mizoram, free of charge to people who tested positive for HIV. Hence, the year 2007 was chosen as the baseline for this study. 

The registration of PWID into TI services was conducted by MSACS through non-governmental organization-supported targeted intervention (TI-NGO). Data were collected by trained peer educators (PEs) and outreach workers (OWs) from 34 TI NGOs in eight (8) districts across Mizoram [13]. Individuals who reported injecting drugs at least once 3 months before the date of data collection were eligible participants for enrolment in TI services. Yearly follow-up of registered PWID was performed to monitor the adherence to services, death rates, and those lost to follow-up. Participants who had died at any time during follow-up between 2007 and 2021 were included in the study. The comprehensive data collection procedures used in this study have been described elsewhere [15].

### 2.2. Outcome Measures

Mortality rates were ascertained through the records from MSACS. The causes of death of PWID were not recorded or classified. In this study, the outcome of interest was mortality among HIV-positive PWID and was coded binary 1 for ‘Yes’ and 0 for ‘No’. The potential confounders that were considered were influenced by a previous similar study [5] and were classified into three main factors, namely, sociodemographic factors, injecting behavior, and sexual behavior. The sociodemographic characteristics included age category (‘18–24’, ‘25–34’, and 35+), gender (male/female), marital status (never married, married, or separated/divorced/widowed), educational status (primary, middle, higher, or graduate and above), employment status (unemployed, employed, or self-employed), and average monthly income in Indian rupees (INR) (None, <3000, 3001–6000, 6001–10,000, or >10,000). Injecting behavior factors included sharing of needles/syringes (Yes/No). Factors related to sexual behavior included whether the person used a condom with a regular partner (Yes/No). 

### 2.3. Statistical Analysis

STATA (Stata Corp., College Station, TX, USA, version 17.0) was used for all analyses. For categorical data, the preliminary analysis conducted has been summarized as total deaths and a total population of each confounding factor. The mortality rates for HIV-negative PWID and HIV-positive PWID were calculated by dividing the total deaths by the total population and multiplying by 1000, as well as their mortality rates and 95% confidence interval (CI) for all potential confounding factors. Using logistic regression models, the univariate analysis examined the independent association between the outcome and the confounding factors, and the multivariable analysis examined the independent risk factors for each study outcome variable after controlling for all potential confounding factors.

In the univariate analysis, all confounding factors with a *p*-value < 0.20 were retained and were used to build a multivariable logistic regression model [16]. A manual backyard elimination procedure was applied for multivariate logistic regression to remove non-significant variables (*p* > 0.05). Only those statistically significantly related to the study outcomes at a 5% significance level in the final model are reported in the study.

## 3. Results

### 3.1. Trends in the Mortality among HIV-Negative PWID and HIV-Positive PWID (2007–2021)

Figure 1 depicts the overall trend in mortality among HIV-negative PWID from the years 2007 to 2021. Overall, there was a decline in the mortality rate per 1000 people among HIV-negative PWID from 89 (95% CI 78–98) in 2007–2011 to 55 (95% CI 46–65) in 2012–2016 and 30 (95% CI 25–35) in 2017–2021. However, there were fluctuations in the mortality rate per 1000 people among HIV-positive PWID. As shown in Figure 2, there was a significant drop from 100 (95% CI 80–120) in 2012–2016 to 58 (95% CI 45–71) in 2017–2021. In the first period, between 2007 and 2011, the mortality rate was 99 (95% CI 73–126).

### 3.2. Mortality Rates among HIV-Negative PWID and HIV-Positive PWID 

Table 1 shows the mortality rates among HIV-negative and HIV-positive PWID by sociodemographic, injecting, and sexual factors. As shown, the mortality rate per 1000 people for PWID was highest for HIV-positive females (115, 95% CI 82–148) followed by HIV-negative males (56, 95% CI 51–61). The mortality rate was high for those aged 24–34 years for both HIV-negative PWID (63, 95% CI 56–70) and HIV-positive PWID (92, 95% CI 76–109). Mortality was also high among divorced/separated/widowed HIV-negative PWID (70, 95% CI 57–83) and HIV-positive PWID (102, 95% CI 79–126). The mortality rate was higher for HIV-negative PWID who shared needles/syringes (63, 95% CI 53–72) and slightly higher for HIV-positive PWID who did not share needles/syringes (81, 95% CI 68–94). Those who did not use condoms with regular partners had higher mortality among the HIV-positive PWID (104, 95% CI 85–124).

### 3.3. Multivariable Analysis of Factors Associated with Mortality among HIV-Positive PWID

Table 2 shows the unadjusted and adjusted odds ratios of factors associated with mortality among HIV-positive PWID. Only factors identified as significant were included in the multivariable analysis. Our analysis showed that mortality among HIV-positive PWID was significantly low between 2017 and 2021 (AOR = 0.57, 95% CI 0.38–0.82). After adjusting for potential confounders, mortality among HIV-positive PWID remained positively associated with being divorced/separated/widowed (AOR = 1.41, 95% CI 1.03–1.94). PWID who used condoms with regular partners had a lower mortality rate (AOR = 0.72, 95% CI 0.55–0.96).

### 3.4. Multivariable Analysis of Factors Associated with Mortality among HIV-Negative PWID

The unadjusted and adjusted odds ratios of factors associated with mortality among HIV-negative PWID are shown in Table 3. Only factors identified as significant were included in the multivariable analysis. Our analysis found that the mortality among HIV-negative PWID was significantly low during the periods of 2012–2016 (AOR = 0.58, 95% CI 0.47–0.72) and 2017–2021 (AOR = 0.24, 95% CI 0.19–0.30). After adjusting for potential confounders, the mortality among HIV-negative PWID remained positively associated with ages of 25–34 years (AOR = 1.54, 95% CI 1.29–1.84) and above 35 years (AOR = 2.08, 95% CI 1.52–2.86). Mortality was also positively associated with being separated/divorced/widowed (AOR = 1.28, 95% CI 1.02–1.61) and sharing of needles/syringes (AOR = 1.64, 95% CI 1.34–2.00). Conversely, the mortality among HIV-negative PWID was negatively associated with being married (AOR = 0.72, 95% CI 0.57–0.90), being employed (AOR = 0.77, 95% CI 0.64–0.94), and having a monthly income between INR 3000 and 10,000.

## 4. Discussion

This study is the first to estimate the mortality rates among PWID and describe the associated factors affecting mortality among HIV-positive and HIV-negative PWID in Mizoram, India. This study showed that there was an overall decline in the mortality rate among HIV-negative PWID from 2007 to 2021. The mortality among HIV-positive PWID remained stable between 2007 and 2016; however, from 2017 to 2021, the mortality among HIV-positive PWID declined by almost fifty percent, while mortality among HIV-negative PWID has reduced significantly by 76% in the past 15 years. Multivariable regression analyses showed that being divorced/separated/widowed had a positive association with mortality among HIV-positive PWID, and those who used condoms with regular partners had a lower mortality rate. The study also found that being between the ages of 24 and 34 years, being above 35 years of age, being separated/divorced/widowed, and sharing of needles/syringes contributed to higher odds of mortality among HIV-negative PWID. We also found that HIV-negative PWID who were employed and had a monthly income had lower odds of mortality.

This study found a noteworthy reduction in mortality rates among PWID over the last 15 years. The results differ from a cohort study conducted in Hai Phong, Vietnam, which examined mortality rates among PWID and found a high death rate among HIV-positive and HIV-negative PWID, with 67 HIV-positive and 36 HIV-negative deaths among 1658 participants over a median follow-up of 2 years [17]. Past studies have shown that drug overdose and liver-related disease [11,17,18] are the underlying causes of death among HIV-negative PWID. Drug overdose is preventable and treatable through the use of naloxone [19,20]. To prevent overdose and its associated harms, including death, the United Nations Office on Drugs and Crime (UNODC) in 2013, in collaboration with the Government of Mizoram, organized training in government hospitals and private institutions on ‘Overdose Management and Prevention’, including assisting in the procurement and distribution of naloxone in district hospitals in Mizoram [19]. The findings in this study suggest that the distribution of naloxone in district hospitals and training in overdose prevention might have prevented deaths from drug overdose among this population. However, decreased mortality among this population cannot be achieved by one treatment modality alone. Comprehensive interventions that include needle/syringe programs, opioid substitution therapy, condom distribution programs, education, and communication are also important strategies that focus on addressing harms associated with drug use [21,22].

The finding that mortality has declined among HIV-positive PWID reported in this study may be attributed to the increased access to antiretroviral treatment (ART), adherence to treatment, and availability of support services. The decline in mortality among this population could be attributed to the launch of free ART in India in 2004 in its fight against HIV/AIDS [23]. Although the initial rollout of ART was slow and limited, the third phase of the National AIDS Control Program (NACP-III), launched in 2009, provided a great impetus to scale up and increase access to services, including ART provision centers [23]. Various studies [24,25,26] have shown that early initiation and adherence to ART have been able to improve the survival of HIV-affected individuals. However, the eligibility for initiation of ART was based on CD4+ count and WHO clinical staging [24,27]. This meant that not all HIV-affected individuals were eligible for free ART, but in 2017, MSACS launched the ‘test and treat strategy’ to improve the treatment of HIV-affected individuals in Mizoram. Under this strategy, people living with HIV were given free ART, irrespective of the CD4+ count [28]. This strategy promoted early initiation of ART [26] and may have led to a decrease in the mortality rate since 2017. In addition to this, the availability of experienced clinicians, ART medication administration, and adequate support services [29] may have enhanced ART adherence among PWID, which in turn may have led to significant reductions in mortality among PWID in Mizoram. The exact effect of ART on mortality among HIV-positive PWID could not be explored in this current analysis. Further research is needed to examine the potential benefits of retention and adherence to ART among HIV-positive PWID on mortality.

Our research showed that the mortality rates for both HIV-negative and HIV-positive PWID were higher among those who were separated/divorced/widowed. Studies have shown that separated/divorced individuals have a wider sexual network, leading to more sexual partners and a greater risk of contracting HIV/AIDS, which can ultimately lead to death [30]. This supports the findings of the National Longitudinal Mortality Study conducted in the United States, which found that individuals who were divorced or separated had a 4.3 times higher risk of dying from HIV/AIDS compared to those who were married [30]. 

Our research has shown that HIV-negative PWID who are between the ages of 24 and 34 years are at greater risk of mortality. Injected drug use is the most common risk factor for drug overdose in young people [31]. A study in San Francisco found high mortality among young PWID (median age: 26 years), and drug overdose was the leading cause of death (57.9%) in this cohort [32]. In addition, our study also found that PWID aged 35 years and older who reported being HIV-negative were more likely to die than those aged 18–24 years, as they may have been exposed to injected drug use for a longer period of time. These findings are consistent with a longitudinal HIV prevention study conducted in Denver, which found that individuals aged over 35 years had a higher risk of mortality compared to those aged 24–35 years [33]. However, our findings contradict the study’s conclusion that those in the age group of 25–34 are at a lower risk of mortality among HIV-negative PWID [33]. 

Our study also reported that sharing needles is a significant predictor of mortality among HIV-negative PWID. A community-based cohort study of PWID conducted in Tijuana [34] found that violence exposure, including interactions with law enforcement, was a significant predictor of mortality among this key population. Fear of police violence and injuries sustained during beatings led to increased substance use to deal with pain, and this contributed to riskier injection behaviors, including the sharing of needles/syringes [34]. Police encounters can also drive mortality among PWID by creating barriers to accessing a range of necessary health and harm reduction services [34,35].

Employment and earning a monthly income were found to be linked to a decreased risk of mortality among HIV-negative PWID in this study. These results align with a previous study by DeBeck et al. [36] on the income-generating activities of people who inject drugs, which showed that earning money could help PWID avoid violence, criminal behavior, and death. This finding shows that having a stable job can reduce involvement in the sex trade industry and high-risk behaviors, such as decreased daily drug use. It also emphasizes the significance of better socioeconomic conditions as a crucial health determinant [17,37].

Despite the reduction in mortality observed in this study, HIV infection continues to play a significant role in the mortality of PWID in Mizoram. The findings of our study highlight the importance of continued promotion and implementation of HIV harm reduction services, including encouraging safe injection practices and engaging in chronic disease management and health promotion activities [5]. Comprehensive training for laypersons and family members of drug users should include how to respond to overdoses, and the administration of naloxone is required to prevent opioid-related drug overdose deaths [38]. There is also a need for the provision of training among police agencies/law enforcement on service delivery and health promotion strategies for PWID, as this can reduce behaviors that interfere with the achievement of public health goals [35]. Structural interventions on the integration of police agencies/law enforcement and public health could serve as an opportunity to connect PWID with support services, which can result in reduced mortality among PWID [34,39]. Perhaps the involvement of PWID in the design and delivery of such services would be an effective community-based health promotion strategy and may address the factors related to employment and skill development.

## 5. Strengths and Limitations

This study has certain strengths. Firstly, this is the first study on the impact of HIV infection and its associated factors on mortality among PWID in Mizoram. The second strength is that this is a population-based study with a large sample size, and we were able to report detailed data for a period of 15 years on this otherwise hard-to-reach population. Our study may also be limited by several factors. Firstly, the study could not determine cause-specific mortality because details on drug-related time of death and chronic disease-related time of death were missing from the MSACS datasets. Secondly, we could not determine how many HIV-positive PWID were enrolled in the ART program or how many PWID were retained due to the limitation of our access to this data. Thirdly, the mortality rates may have been underestimated, as participants who were registered in the TI program but discontinued the service may have died but were not followed up.

## 6. Conclusions

In conclusion, our results demonstrate that mortality rates among HIV-negative and HIV-positive PWID have declined significantly since 2016. This coincided with the launch of the ‘test and treat strategy’ and ‘overdose management and treatment’ in Mizoram at the time. However, this finding does not negate the need to continue a comprehensive response, including a scale-up of ART, education of laymen on overdose management and safe injection practices, and the involvement of law enforcement in public health goals for PWID to further reduce mortality among this vulnerable group. Our study findings also suggest that employment and income generation among PWID could help to avoid violence and death. Public health responses need to include skill development for PWID as an intervention strategy to reduce mortality among this population.

## Figures and Tables

**Figure 1 ijerph-21-00874-f001:**
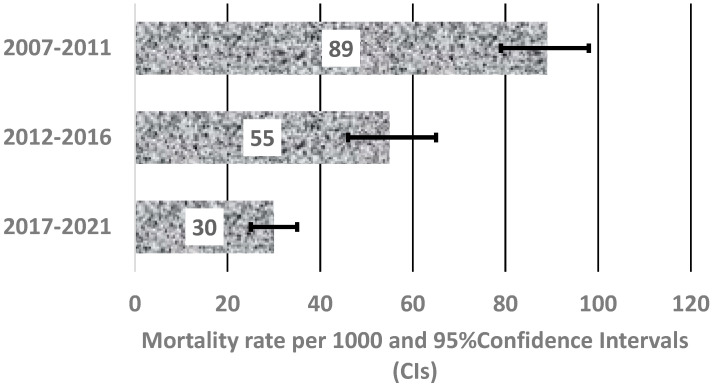
Mortality rate per 1000 people and 95% CIs among HIV-negative PWID.

**Figure 2 ijerph-21-00874-f002:**
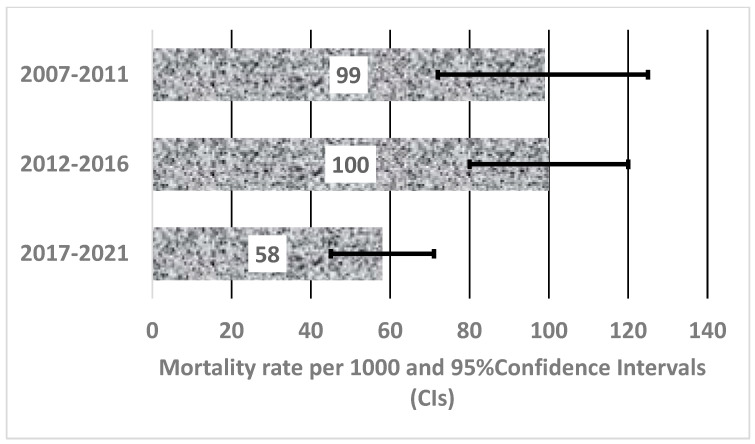
Mortality rate per 1000 people and 95% CIs among HIV-positive PWID.

**Table 1 ijerph-21-00874-t001:** HIV-negative PWID and HIV-positive PWID mortality rates with 95% confidence intervals.

Covariates	HIV-Negative PWID			HIV-Positive PWID		
	Total (n)	No. of Deaths	MR (95% CI)	Total (n)	No. of Deaths	MR (95% CI)
HIV status by year of registration						
2007–2011	3976	352	89 (79–98)	612	61	99 (73–126)
2012–2016	2465	136	55 (46–65)	1044	105	100 (80–120)
2017–2021	5104	152	30 (25–35)	1480	86	58 (45–71)
**Gender**						
Male	10,886	610	56 (51–61)	2675	199	74 (64–85)
Female	658	30	46 (29–62)	461	53	115 (82–148)
**Age**						
18–24 years	5359	258	48 (42–54)	1339	93	69 (55–84)
25–34 years	4943	311	63 (56–70)	1430	132	92 (76–109)
35+ years	999	57	57 (41–72)	294	20	68 (37–98)
**Marital status**						
Not married	7219	404	56 (50–62)	1704	119	70 (57–83)
Married	2625	113	43 (35–51)	624	44	71 (49–92)
Divorced/separated/widowed	1676	117	70 (57–83)	791	81	102 (79–126)
**Education status**						
Primary (0–6 years)	1632	104	64 (51–76)	353	35	99 (65–134)
Middle (7–9 years)	3896	243	62 (54–70)	1102	109	99 (79–118)
Higher (10–12 years)	5564	270	49 (43–54)	1559	102	65 (52–79)
Graduate and above	431	20	46 (26–67)	118	6	51 (9–92)
**Employment status**						
Unemployed	6195	396	64 (57–70)	1438	108	75 (60–90)
Employed	4040	198	49 (42–56)	1350	114	84 (68–101)
Self- employed	1308	46	35 (25–46)	346	30	87 (54–119)
**Average monthly income (INR)**						
None	4432	344	78 (69–86)	879	75	85 (65–105)
<3000	3186	129	40 (33–48)	1189	93	78 (62–95)
3001–6000	2493	120	48 (39–57)	628	55	88 (63–112)
6001–10,000	928	20	22 (12–31)	270	20	74 (40–108)
>10,000	451	25	56 (33–78)	119	7	59 (14–104)
**Sharing of needles/syringes**						
No	8467	450	53 (48–58)	2016	164	81 (68–94)
Yes	3021	190	63 (53–72)	1106	88	80 (62–97)
**Condom use with regular partner**						
No	3393	181	53 (45–61)	1149	120	104 (85–124)
Yes	7334	418	57 (51–63)	1572	110	70 (56–84)

MR = Mortality rate per 1000 people. This table presents the mortality rates among PWID by sociodemographic, injecting, and sexual factors.

**Table 2 ijerph-21-00874-t002:** Unadjusted and adjusted odds ratios of factors associated with mortality among HIV-positive PWID.

Characteristics	OR (95% CI)	*p*-Value	AOR (95% CI)	*p*-Value
HIV status by year of registration				
2007–2011	1		1	1
2012–2016	1.01 (0.72–1.14)	0.953	0.99 (0.69–1.42)	0.976
2017–2021	0.55 (0.39–0.78)	0.001	0.57 (0.38–0.82)	0.002
Gender				
Male	1			
Female	1.61 (1.17–2.22)	0.003		
Age				
18–24	1			
25–34	1.36 (1.03–1.79)	0.028		
>35	0.97 (0.59–1.61)	0.930		
Marital status				
Never married	1		1	
Married	1.01 (0.71–1.44)	0.955	1.04 (0.72–1.51)	0.832
Separated/divorced/widowed	1.52 (1.13–2.04)	0.006	1.41 (1.03–1.94)	0.032
Education status				
Primary (0–6 years)	1			
Middle (7–9 years)	0.99 (0.66–1.48)	0.990		
Higher (10–12 years)	0.64 (0.42–0.95)	0.028		
Graduate and above	0.49 (0.20–1.18)	0.114		
Employment status				
Unemployed	1			
Employed	1.13 (0.86–1.49)	0.363		
Self employed	1.16 (0.76–1.78)	0.469		
Average monthly income (INR)				
None	1			
<3000	0.91 (0.66–1.24)	0.559		
3001–6000	1.02 (0.71–1.48)	0.878		
6001–10,000	0.85 (0.51–1.43)	0.557		
>10,000	0.62 (0.28–1.37)	0.232		
Sharing of needles/syringes				
No	1			
Yes	0.97 (0.75–1.28)	0.861		
Condom use with regular partner				
No	1		1	1
Yes	0.65 (0.49–0.85)	0.002	0.72 (0.55–0.96)	0.025

This table presents the sociodemographic, injecting, and sexual risk factors associated with mortality among HIV-positive PWID before and after adjusting for potential confounders.

**Table 3 ijerph-21-00874-t003:** Unadjusted and adjusted odds ratios of factors associated with mortality among HIV-negative PWID.

Characteristics	OR (95% CI)	*p*-Value	AOR (95% CI)	*p*-Value
**HIV status**				
2007–2011	1		1	
2012–2016	0.60 (0.49–0.74)	<0.001	0.58 (0.47–0.72)	<0.001
2017–2021	0.31 (0.26–0.38)	<0.001	0.24 (0.19–0.30)	<0.001
**Gender**				
Male	1			
Female	0.80 (0.55–1.17)	0.257		
**Age**				
18–24	1		1	
25–34	1.31 (1.12–1.57)	0.001	1.54 (1.29–1.84)	<0.001
>35	1.19 (0.89–1.61)	0.234	2.08 (1.52–2.86)	<0.001
**Marital status**				
Never married	1		1	
Married	0.75 (0.61–0.93)	0.011	0.72 (0.57–0.90)	0.005
Separated/divorced/widowed	1.26 (1.02–1.57)	0.030	1.28 (1.02–1.61)	0.033
**Education status**				
Primary (0–6 years)	1			
Middle (7–9 years)	0.97 (0.77–1.23)	0.850		
Higher (10–12 years)	0.75 (0.59–0.94)	0.015		
Graduate and above	0.71 (0.44–1.17)	0.180		
**Employment status**				
Unemployed	1		1	
Employed	0.75 (0.63–0.89)	0.002	0.77 (0.64–0.94)	0.012
Self employed	0.53 (0.39–0.73)	<0.001	0.78 (0.56–1.10)	0.17
**Average monthly income (INR)**				
None	1		1	
<3000	0.50 (0.41–0.62)	<0.001	0.58 (0.46–0.72)	<0.001
3001–6000	0.60 (0.48–0.74)	<0.001	0.66 (0.52–0.84)	0.001
6001–10,000	0.26 (0.16–0.41)	<0.001	0.33 (0.20–0.55)	<0.001
>10,000	0.69 (0.46–1.05)	0.091	1.08 (0.67–1.72)	0.739
**Sharing of needles/syringes**				
No	1		1	
Yes	1.19 (1.0–1.42)	0.045	1.64 (1.34–2.00)	<0.001
**Condom use with regular partner**				
No	1			
Yes	1.07 (0.89–1.28)	0.444		

This table presents the sociodemographic, injecting, and sexual risk factors associated with mortality among HIV-positive PWID before and after adjusting for potential confounders.

## Data Availability

Data are available upon request.

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
