# Peer review of "Mortality among HIV-Positive and HIV-Negative People Who Inject Drugs in Mizoram, Northeast India: A Repeated Cross-Sectional Study (2007–2021)"

_ijerph, 2024, doi:10.3390/ijerph21070874_

Round 1
Reviewer 1 Report
Comments and Suggestions for Authors
This manuscript reported the trends of death rate and related factor in injecting drug users with and without HIV infection in Mizoram, India. The results were informative and attractive. However, I have some concerns.
1. The study design: the authors described the study samples and design in the methods (lines 92-112). However, the design was not clear. The authors wrote that registered PWID were yearly followed up. So, I think the data can be analyzed as longitudinal study (such as Kaplan-Meier and Cox regression). Then, the results would be much easier to compare between the HIV-infected and -uninfected participants, even with other reports (such as reference 11).
2. In the introduction, the authors introduced the mortality rate in PWID in Chennai, and AIDS-related death rate in India including Mizoram. I want to know what the level of the PWID death rates in this report. Are the rates higher or lower than that in local general population? Moreover, in the discussion, (line 230-234), the authors described that the authors’ results differed from a study (reference 17) which found a high death among HIV-negative PWID. Here, the author did not show how differ between these two studies? The death rates differ? Your reduction of mortality rates differs? Or your low death rate in HIV(-) differs? By the way, reference 17 did not report high death rate in HIV(-), their death rates were 1.9/100 person-years in HIV(-) vs. 4.3/100 person-years in HIV(+).
3. Results 3.2, Tables 1 shows the comparison of mortality rates… But I could not see any comparison in the table. Moreover, the factors (or characteristics) are not covariate here because the authors did not do multiple statistical analysis in this table.
4. Table 2 and 3, HIV status seems not be appropriate. Instead, period or duration could be a choice.
5. Lines 281-288, the authors wrote that HV-negative PWID who are between the ages of 24-34 years are at greater risk of mortality. In addition, individuals aged 35 years and older were more likely to die than those aged 18-24 years as they have been exposed to injecting drug use for a longer period of time [17]. First, reference 17 did not show this information, whereas it showed that only age groups 48-74 had significantly higher AHR than group 19-35 [2.5 (1.4-4.5)], though crude HR of 43-48 group was significant. Moreover, in this report, the group >35 had higher AOR than group 25-34 (2.08 vs. 1.54), which may indicate group>35 had higher risk than group 25-34? By the way, the tense of English in this paragraph is a bit confusing.
The authors described that they did not analyze the death factors in this study, so the discussion was mainly based on others’ reports which made the paper a littler weak. For example, lines 289-295, which discussed the possible reason of sharing needle and violence exposure. The discussion was totally based on estimation because the authors had not information for their participants. Moreover, I wonder why the sharing needle was not related with death in HIV (+). Did HIV(+) PWID had less violence exposure than HIV(-) PWID? Thus, I strongly recommend the authors to include some death reasons in the analysis if possible.
Comments on the Quality of English LanguageEnglish is good.
Reviewer 2 Report
Comments and Suggestions for Authors
IJERPH/ijerph-3040884-peer-review-comments-v1
Trends of HIV mortality rate and its associated factors among people who inject drugs in Mizoram, Northeast India.
Lucy Ngaihbanglovi Pachuau et.al.
Overall Comments :
The paper addresses an important subject and is well-written. However, it has some significant limitations. There is a lack of information on important parameters related to cause-specific mortality, such as drug-related deaths, chronic disease-related deaths, data on the enrollment of HIV-positive PWID in the ART Programme, and the retention of PWID in the program. Additionally, there is an underestimation of mortality rates due to the lack of follow-up and the likelihood of death among participants who were registered in the TI Programme but discontinued the service.
The absence of such data could result in biased and misleading conclusions in some instances. Despite these limitations, the paper may be considered for publication after a thorough review and revision, especially considering that it is supposed to be the first published report in Mizoram as stated by the authors.
Specific Comments :
Abstract:
Methods: Line 23- It is important to review and revise this sentence to ensure clarity and proper conveyance of meaning.
Introduction:
Please ensure that lines 80, 81, 84, and 85 convey the intended meaning accurately. In lines 84 and 85, it's important to correctly reflect the study's objective. Additionally, it would be beneficial to include information about current intervention strategies for reducing HIV-related deaths among people who inject drugs and the existing data gap in Mizoram. Some relevant information has already been included under the Material and Methods section (2.1 Study sample and design). Consider moving this information to the Introduction section for better clarity to the readers.
Discussion :
The authors published another paper of a similar nature on 22 May 2023 in PLOSONE (URL_1). I did not find any reference to this paper. Please discuss how the findings of this paper differ from or contribute to the existing knowledge on the subject addressed in your current paper submitted for publication. Additionally, there are similar studies from certain Indian states, such as Manipur, which could be included in the discussion.
The limitations of this study are significant. For example, there are challenges in determining cause-specific mortality, such as drug-related deaths and chronic disease-related deaths. There is also a lack of data on how many HIV-positive people who inject drugs (PWID) were enrolled in the Antiretroviral Therapy (ART) Programme, as well as how many PWID were retained in the program. Additionally, there is a concern about underestimating mortality rates due to the lack of follow-up and the likelihood of death of participants who were registered in the Targeted Intervention (TI) Programme but discontinued the service. It is unclear whether these limitations are due to the non-availability of the data with the State of Mizoram or the authors' access to this data. Please provide clarification on this matter.
Conclusion :
Lines 336-342: It appears that the conclusions drawn in this study are not supported by the results. It would be advisable to revise them in accordance with the paper's objectives. Any recommendations should be based on the findings of this study and included at the end.
References :
Review to ensure uniform journal format across all references and add DOIs to keep up with modern publication trends.
Comments on the Quality of English Language
Minor editing and revision for clarity regarding the intended meaning are required.
Reviewer 3 Report
Comments and Suggestions for Authors
Below are my comments on the manuscript:
Line 45: issing a space
Line 95: Initially, who fills out or completes the information for each PWID: the doctor, a social worker or the PWID themselves? My question is about the possibility of including biases in the information.
Line 106: Do you have information about the month of ART introduction??? I mean not only the year.
Line 112: What was the criteria for choosing 3 months of injecting drugs?
Line 129: Is there information about casual couples and condom use?
Line 145: -missing space/-what was the criteria to establish the cut-off point of 0.20?
Line 151: I do not know if it is a journal requirement not to include figure o table captions/footnote. If not, I recommend that a figure/table caption accompany these elements to give a brief explanation of what they show.
Line 273: This is one of the reasons why I question sexual behavior and condom use with casual partners.
